# Polyphenols Content in *Capsicum chinense* Fruits at Different Harvest Times and Their Correlation with the Antioxidant Activity

**DOI:** 10.3390/plants9101394

**Published:** 2020-10-20

**Authors:** Julio Enrique Oney-Montalvo, Kevin Alejandro Avilés-Betanzos, Emmanuel de Jesús Ramírez-Rivera, Manuel Octavio Ramírez-Sucre, Ingrid Mayanin Rodríguez-Buenfil

**Affiliations:** 1Centro de Investigación y Asistencia en Tecnología y Diseño del Estado de Jalisco A.C. Sede Sureste, Tablaje Catastral 31264 Km, 5.5 Carretera Sierra Papacal-Chuburna Puerto Parque Científico Tecnológico de Yucatán, Mérida 97302, Mexico; juoney_al@ciatej.edu.mx (J.E.O.-M.); keaviles_al@ciatej.edu.mx (K.A.A.-B.); emramirez_al@ciatej.edu.mx (E.d.J.R.-R.); oramirez@ciatej.mx (M.O.R.-S.); 2Tecnológico Nacional de México/Tecnológico Superior de Zongolica, Km, 4 Carretera S/N Tepetlitlanapa, Zongolica 95005, Veracruz, Mexico

**Keywords:** flavonoids, phenolic acids, Habanero pepper, linear correlation, maturity, UPLC-DAD

## Abstract

The aim of this work was to investigate the changes of the content of polyphenols in fruits of *Capsicum chinense* Jacq. at different harvest times and their correlation with the antioxidant activity. Habanero pepper plants grown in black soil (Mayan name: *Box lu’um*) and harvested at 160, 209, 223, 237 and 252 post-transplant days (PTD) were analyzed. The results indicated that subsequent harvesting cycles decreased the content of total polyphenols, catechin, chlorogenic acid and ellagic acid, while the content of gallic and protocatechuic acid increased. The antioxidant activity determined by DPPH (2,2-Diphenyl-1-picrylhydrazyl) radical scavenging and ABTS (2,2′-azino-di-3-ethylbenzthiazoline sulfonic acid) assay decreased through the harvest days. Linear correlation analysis between total polyphenol content and antioxidant activity in peppers resulted in a correlation of r^2^_DPPH_ = 0.8999 and r^2^_ABTS_ = 0.8922. Additionally, a good correlation of the antioxidant activity was found with catechin (r^2^_DPPH_ = 0.8661 and r^2^_ABTS_ = 0.8989), chlorogenic acid (r^2^_DPPH_ = 0.8794 and r^2^_ABTS_ = 0.8934) and ellagic acid (r^2^_DPPH_ = 0.8979 and r^2^_ABTS_ = 0.9474), indicating that these polyphenols highly contributed to the antioxidant activity in Habanero peppers. This work contributes to understanding the changes that take place during the development of *Capsicum chinense*, indicating that fruit harvested at earlier PTD showed the highest concentrations of total polyphenols and antioxidant activity, obtaining the best results at 160 PTD.

## 1. Introduction

Mexico is considered one of the main producers and exporters of peppers in the world. The peppers are products with a high socio-economic impact in the country, particularly in employment generation, foreign exchange earnings and agricultural value chains [1]. Among the peppers grown in Mexico, the Habanero pepper (*Capsicum chinense* Jacq.) of the Yucatan Peninsula is considered one of the most important [2,3]. This is due to its high content of capsaicinoids, which classifies it as one of the hottest peppers in the world [4,5]. It has also been recognized nationally and internationally by the designation of origin obtained in 2010 (“Chile Habanero de la peninsula de Yucatán”) by the “Mexican Institute of Intellectual Property” [6,7].

The Habanero pepper is an annual cycle plant that reaches a height of 1.5 m and lives up to 16 months; of all varieties, this pepper develops the highest intensity of spicy flavor in the entire *Capsicum* genus [8]. Immature specimens of the Habanero pepper are of a green color, but this color varies with maturity (until it obtains an orange color); the fruit height also depends on the maturity stage and is between 2 and 6 cm in size with a pungency between 100,000–300,000 Scoville units [9].

The Habanero pepper is characterized by the presence of different secondary metabolites of great importance for the food and pharmaceutical industry; these compounds are: capsaicinoids, polyphenols, vitamins and carotenoids [10]. Polyphenols are one of the main metabolites as well as one of the most studied in recent years [11,12]. The presence of these compounds in Habanero pepper provides added value and offers health benefits to the consumer as the prevention of chronic degenerative diseases, for example: cardiovascular and neurogenerative diseases [10]. Previous work conducted by Dubey et al. [13] determined catechin (18.03 mg/100 g of dry pepper) as the main polyphenol in peppers, followed by quercetin (3.87 mg/100 g of dry pepper), protocatechuic acid (2.06 mg/100 g of dry pepper) and rutin (0.82 mg/100 g of dry pepper). These results coincided with those reported by Troconis-Torres et al. [14], who quantified catechin (52.25 mg/100 g of dry pepper) as the major polyphenol in Habanero pepper.

The polyphenol content in Habanero pepper is developed due to different factors, the main ones are the genotype of the plant [15], degree of maturity of the fruit [16], physicochemical characteristics of the soil where the plant grown and environmental conditions [17,18]. The cultivation cycle of the Habanero pepper is also a factor that affects the production and quality of the fruit [19]. The life cycle of the Habanero pepper plant mainly comprises four phenological stages: (1) vegetative, (2) flowering, (3) fruiting and (4) production. Vegetative stage occurs approximately at 50 post-transplant days (PTD). The flowering stage normally happens at 75 PTD, the fruiting stage starts at 100 PTD and the production stage starts approximately at 120 PTD [20]. The work carried out by Bhandari et al. [21] showed that the concentration of capsaicinoids, ascorbic acid, total polyphenols and β-carotene in the Habanero pepper changed throughout the production stage (some metabolites increased, while others decreased). This phenomenon may be a result of changes in the biosynthesis, translocation or degradation of these metabolites [21].

The same behavior has been reported on the antioxidant activity in peppers, which decreased throughout the harvest period [21]. This is mainly related to the change in the content of flavonoids and phenolic acids that have shown to play an important role in antioxidant activity, reporting linear correlations of up to 0.7 between total polyphenol content and antioxidant activity in red pepper (*Capsicum annuum* L.) [21]. The antioxidant activity of polyphenols is mainly caused by the chemical structure of these molecules, as the number and position of hydroxyl groups provide the antioxidant activity of the molecule [22]. Nevertheless, it is also inversely proportional to the enthalpy of dissociation of the hydrogen and oxygen atoms bond of the hydroxyl group [23].

Changes have been observed in the concentration of secondary metabolites in the fruit during the production stage of the Habanero pepper plant. However, how this factor affects the production of specific polyphenols such as flavonoids and phenolic acids and the antioxidant activity has not been reported yet. Due to the above, the objective of this work was to evaluate the content of 19 individual polyphenols, as well as the total polyphenols in the *Capsicum chinense* fruits at different harvest times, and analyze the correlations with antioxidant activity.

## 2. Results

### 2.1. Quantification of Polyphenols

Results of quantification of polyphenols on Habanero pepper with two grades of maturity (mature and immature) and different harvest time (PTD) by Ultra Performance Liquid Chromatography (UPLC) are presented in Table 1. It can be observed that the concentration of polyphenols changes at different harvest times and by the degree of maturity. The mature peppers harvested at 160 PTD showed the highest concentration of catechin (355.30 ± 5.81 mg/100 g), chlorogenic acid (79.97 ± 2.02 mg/100 g), coumaric acid (2.31 ± 0.54 mg/100 g), cinnamic acid (25.78 ± 5.14 mg/100 g), diosmin + hesperidin (14.28 ± 0.07 mg/100 g), neohesperidin (2.65 ± 0.03 mg/100 g), apigenin (2.05 ± 0.08 mg/100 g), vanillic acid (45.48 ± 0.55 mg/100 g), ferulic acid (12.12 ± 0.05 mg/100 g) and ellagic acid (7.78 ± 0.03 mg/100 g). On the other hand, mature peppers harvested at 252 PTD had the highest concentration of gallic acid (48.40 ± 12.31 mg/100 g), protocatechuic acid (162.21 ± 0.55 mg/100 g) and cinnamic acid (43.41 ± 4.86 mg/100 g). Only vanillin showed the highest concentration (5.86 ± 0.05) in immature peppers harvested at 160 PTD. The polyphenols quercetin and luteolin were determined together because the separation of the peak by UPLC was not obvious; the same happened with diosmin and hesperidin.

Figure 1 shows the chromatograms obtained from the analysis of polyphenol standards, as well as the samples of mature and immature Habanero peppers. In these, an adequate separation of the peaks can be observed, corresponding to the polyphenols determined in the present work. Gallic acid (0.7 min) was the first compound to elute, followed by protocatechuic acid (1.5 min). Most of the polyphenols (catechine, vanillic acid, chlorogenic acid, vanillin, coumaric acid, ferulic acid, cinnamic acid, ellagic acid, rutin) eluted in a time of 3.5 to 6.5 min. On the other hand, diosmin + hesperidin, neohesperidin and luteolin + quercetin had a retention time of 7.3, 7.6 and 8.3 min, respectively. The last compounds that eluted were naringenin, apigenin, kaempferol and diosmetin (From 9 to 10 min). The chromatogram of the mature Habanero peppers was characterized by the peaks corresponding to the polyphenols analyzed presented a higher signal intensity compared to the chromatogram of the immature Habanero pepper. The only exception to this was peak 6, which corresponds to vanillin. This reinforces that the highest concentration of polyphenols at 160 PTD was quantified in mature peppers.

### 2.2. Statistical Analysis of Polyphenols

*P* values obtained from analysis of variance (ANOVA) are shown in Table 2. The results indicate that the grade of maturity, the harvest time (PTD) and the interaction between these factors have a significant effect in almost all the polyphenols evaluated. Only rutin did not show a significant effect with any factor. Moreover, luteolin, quercetin and neohesperidin only had a significant effect by the grade of maturity.

### 2.3. Linear Correlation Analysis

The results of the correlation analysis of the concentration of polyphenols with the antioxidant activity determined by DPPH radical scavenging and ABTS assay in Habanero pepper are shown in Table 3. Mature peppers presented the best linear correlations compared to immature peppers. Otherwise, the determination of the antioxidant activity by ABTS showed the best correlations compared to the DPPH method. Catechin and vanillic acid were the only individual polyphenol with a correlation with r^2^ > 0.7 in immature and mature Habanero peppers in both antioxidant activity methods; this same tendency was observed with correlations in total polyphenols (r^2^ > 0.84). Correlation of chlorogenic acid in mature peppers presented a lightly better fit with antioxidant activity determine by ABTS assay (r^2^ = 0.8934) than the presented by DPPH radical scavenging (r^2^ = 0.8976). Ellagic acid showed similar behavior, reporting a r^2^ = 0.8976 by DPPH radical scavenging and a r^2^ = 0.9474 by ABTS assay. On the other hand, gallic acid, protocatechuic acid, rutin, luteolin, quercetin, kaempferol, neohesperidin, naringenin, apigenin and diosmetin showed a r^2^ < 0.7 under the evaluated conditions.

### 2.4. Polyphenols Change through Harvests

The Figure 2 shows the change in the percentage of polyphenols between the first (161 PTD) and the last harvest (252 PTD), where most of the polyphenols decreased during the harvest period. Figure 2A,B show polyphenols that increased during harvests from immature and mature peppers, respectively; of these, gallic and protocatechuic acid augmented in mature and immature peppers. All the polyphenols that increased during harvests presented a non-well fitted correlation (r^2^ < 0.7) with the antioxidant activity.

On the other hand, polyphenols showed a good correlation (r^2^ > 0.7), and the major ones (catechin, chlorogenic acid and vanillic acid) showed a decreased greater than 80% through the harvest period. This effect was observed in both grades of maturity of peppers (mature and immature). In Figure 2C, polyphenols from immature peppers that decreased during harvests are presented, while Figure 2D shows the polyphenols from mature peppers that showed a decrease in the percentage through the harvest days.

## 3. Discussion

The catechin was reported as the major polyphenol by Troconis-Torres et al. [14] with a concentration of 52.25 mg/100 g; this result coincided with the research done by Dubey et al. [13], who quantified phytochemical composition in indigenous peppers from India, finding catechin as the highest polyphenol in all the peppers analyzed with a range of concentration of 2.79 to 18.03 mg/100 g. These previous works with the results obtained of the quantification of polyphenols by UPLC in the present work indicate that catechin is one of the major flavonoids in *Capsicum annuum* and *Capsicum chinense*. This tendency was followed by other polyphenols, such as chlorogenic acid (45.72 mg/100 g) and rutin (29.54 mg/100 g), which have been reported in *Capsicum chinense* by Sherova et al. [24]. This tendency was corroborated in this work, but through harvest days, and their correlation was evaluated with the change of the antioxidant activity.

The effect of the degree of maturity on the concentration of polyphenols has been previously studied by Howard et al. [25] and Oney-Montalvo et al. [26], reporting positive changes in the concentration of these compounds during the maturation process of the Habanero pepper. This same increase was observed in the antioxidant activity measured by DPPH and ABTS assay, a phenomenon previously reported by Ghasemnezhad et al. [27]; they found that the antioxidant activity increase with the maturation of bell pepper (*Capsicum annum*). These results can be associated with an increase in the biosynthesis of polyphenols caused by the process of maturation and by the accumulation of nitrate (NO_3_^−1^) and phosphate (PO_4_^−3^) ions. These nutrients play an important role in the biosynthesis of organic compounds, the concentration of these nutrients in the pepper increases through a maturation process, contributing to the production of polyphenols [20,28]. Moreover, Inui et al. [29] studied the effect of the harvest time (PTD) on the concentration of polyphenols and the antioxidant activity and showed that the quantity of polyphenols decreases with the time of harvest and affects the functional properties, including the antioxidant activity. Gao et al. [30] also compared the concentration of polyphenols (caffeic acids, p-coumaric acid, ferulic acid, gallic acid, protocatechuic acid, p-hydroxybenzonic acid and chlorogenic acid) with the antioxidant activity determined by DPPH (2,2-Diphenyl-1-picrylhydrazyl), FRAP (Fluorescence recovery after photobleaching) and TEAC (Trolox Equivalent Antioxidant Capacity) methodology in *Sphallerocarpus gracilis* at two different harvest times, finding that the concentration of polyphenols and antioxidant activity decreased by each harvest time evaluated. This could be caused by the decrease in the concentration of polyphenols throughout the production stage, resulting in changes in the biosynthesis, translocation or degradation of these metabolites [21].

A previous study conducted by Bhandari et al. [21] evaluated the correlation between variations of phytochemicals (flavonoids, vitamins and carotenoids) with antioxidant activity in red peppers (*Capsicum annuum* L.) from South Korea at different harvest times, finding that total flavonoids showed the best correlation (r^2^ = 0.841) of all the metabolites evaluated. In addition, the work done by Mihai et al. [31] in propolis from Transylvania obtained a good fit (r^2^ = 0.8387) of the linear correlation between DPPH values and total polyphenols. These previous works and the results of the present research support that the antioxidant activity is strongly influenced by the presence of polyphenols in the peppers. In Habanero peppers, catechin had a good linear correlation (r^2^ > 0.7) for both maturity stages (immature and mature) with the antioxidant activity determined by DPPH and ABTS. Catechin was also identified as the major polyphenol. Chlorogenic acid was the second highest polyphenol, which influenced the antioxidant activity with an r^2^ > 0.8 for mature peppers. Despite the fact that ellagic acid presented a lower concentration than the previously mentioned polyphenols, it presented the best linear correlation with the antioxidant activity measured by ABTS in mature peppers (r^2^ = 0.9474), indicating that it also contributes to the antioxidant activity of the Habanero pepper. 

The catechin and chlorogenic acid are considered polyphenols that have a strong influence in the antioxidant activity of different foods. In the work conducted by Reddivari et al. [32], these compounds contributed to the antioxidant activity, mainly in potato selections. Moreover, Zapata et al. [33] identified catechin as one of the main polyphenols that increased the antioxidant activity of cocoa beans. On the other hand, ellagic acid is considered a molecule with a strong antioxidant activity, according to the work carried out by Festa et al. [34], who evaluated the scavenging action through their ability to modulate DNA damage produced by two strong radical oxygen inducers (H_2_O_2_ and Bleomycin) in mammalian cells in vitro. Structurally, the antioxidant activity of these compounds may be due to the hydroxyl groups attached to the aromatic rings, which have been related to an increase in antioxidant activity. The hydroxyl groups confer the ability to inhibit the activity of enzymes, chelate ions of metals involved in the process of free radical creation and interrupt the cascade of reactions leading the peroxidation of lipids [23]. The research carried out by Christensen et al. [35] demonstrated that the harvest time has a negative effect on the content of flavonoids and phenolic acids in *Fagopyrum esculentum* and *Fagopyrum buckwheat*, observing a reduction in the concentration of total flavonoids as the harvest dates advance through the crop life cycle. The behavior described above may be due to a decrease in the content of nitrogen and phosphorus in the plant and/or soil, because these are structural nutrients that participate in the synthesis of chemical compounds, such as polyphenols [20].

These results could be caused by a decrease of the nutrients present in the soil. The work carried out by Muscolo et al. [36] has shown that the concentration of polyphenols and antioxidant activity are influenced by the physicochemical composition of the soil. The decrease of nutrients could be occasioned by the demand of nutrients (nitrogen and potassium) and is more pronounced in the phenological stages of fruiting and production, inducing an exhaustion of these chemicals in the soil used for cultivation [20]. Only gallic acid and protocatechuic acid presented an increasing value in immature and mature Habanero peppers through harvests; this could be by a stress caused by a decrease of the available nutrients in the soil that changes the expression of the genes involved in the synthesis of these polyphenols [37].

## 4. Materials and Methods

### 4.1. Plant Growth Conditions

The crop of Habanero peppers (*Capsicum chinense* Jacq. ‘Jaguar’) was established on 14 March 2018. The dates of harvest were selected based on the availability of peppers (>100 for each degree of maturity) grown with two degrees of maturity. The harvests were conducted on 160, 209, 223, 237 and 252 post-transplant days (PTD), selected for the availability of Habanero peppers (>100 for each degree of maturity). The plants were developed in a greenhouse in Sierra Papacal, Yucatán in Mexico (CIATEJ, Sede Sureste). The greenhouse had a north-south orientation, a ridge height of 7.0 m, with a triple-layer plastic cover (25% shade), and lateral walls of high-density plastic anti-trips insect screens. The sample was composed by 100 polyethylene bags, filled with 12 kg of dry black soil (Mayan name: *Box lu’um*). The selection of the soil was due to the research previously done by Oney-Montalvo et al. [24], who showed that the black soil obtained the best results in the production of polyphenols and the antioxidant activity, associating this to the concentration of organic matter (10.93 ± 0.23%), nitrogen (52.01 ± 7.05 mg kg^−1^) and electric conductivity (2.32 ± 0.16 d Sm^−1^). Other physicochemical characteristics of the black soil are: potassium (387.07 ± 4.34 mg kg^−1^), calcium (1823.9 ± 54.22 mg kg^−1^), sodium (11.04 ± 0.85 mg kg^−1^), phosphorous (8.89 ± 0.79 mg kg^−1^), sands (58.20 ± 4.00%), clays (19.85 ± 5.03%) and silts (21.95 ± 7.57%).

Water from a local well was used for irrigation. The electrical conductivity of the water oscillated from 2.8 to 3.4 mS. For fertilization, the methodology of Medina-Lara et al. [38] was used, which is recommended for Habanero pepper cultivated on the soils of Yucatan, with a formula of 120N-100P-150 K kg·ha^−1^ (nitrogen-phosphorus-potassium). The fertilizer used after 10 post-transplant days (PTD) was the Triple 18 Ultrasol^®^ (SQM, Santiago de Chile, Chile) composed of nitrogen, phosphorus and potassium at a concentration of 18%; this was applied in the irrigation with water twice a week. The micronutrients were sprayed on the leaves with commercial product Bayfolan^®^ Forte (Bayer CropScience, Mexico city, Mexico) diluting 24 mL in 16 L of water and applied once a week. After 20 PTD and before floral initiation, a growth regulator containing gibberellin, cytokinin and auxin (Biozyme^®^-TF, Arysta LifeScience, Guatemala city, Guatemala) was applied (16 mL diluted in 16 L of water) once a week. Irrigation was applied twice a week during the first 15 days after PTD; subsequently, the irrigation frequency was maintained at 2 L per polyethylene bag, every third day. Light, thermal and humidity conditions were measured with the help of data loggers randomly located in the greenhouse. The results obtained by the data loggers are presented in the Table A1 of the Appendix A.

### 4.2. Sample Collection and Processing

The peppers were collected and classified by degree of maturity: immature (color green) and mature (color orange) at different harvest times (160, 209, 223, 237 and 252 PTD). Then, approximately 100 peppers for each degree of maturity and harvest time were dried according to the methodology reported by Zamacona et al. [39] in an oven Felisa FE-292 with gravity convection at 65 °C for 72 h. After, the peppers were ground in a mortar until a fine powder was obtained, which was sieved on a 500 µm mesh pore sieve (obtaining approximately 20 g of powder per 100 Habanero peppers). The resulting powder was stored in a polythene bag at room temperature and protected from light until further analysis. Three samples of peppers were analyzed by each combination of maturity stage and planting date. The physical characteristics of the habanero peppers (weight, length and width) are represented in Table A2 located in Appendix B.

### 4.3. Extraction of Habanero Pepper Powder

For the extraction of polyphenols, approximately 500 mg of Habanero pepper powder were mixed with 2.5 mL of MeOH:H_2_O (80:20) solution. The mixture was sonicated at 42 kHz for 30 min at room temperature and then centrifuged at 4700 rpm and 4 °C during 30 min. The supernatant was filtered through a 0.2 µm polytetrafluoroethylene (PTFE) filter and immediately analyzed.

### 4.4. Antioxidant Activity by DPPH Radical Scavenging

The antioxidant activity was determined by 2,2,1-diphenyl-1-picrylhydrazyl (DPPH) radical scavenging method according to Brand-Williams et al. [40] with some modifications. The DPPH solution was prepared in MeOH and diluted to a concentration with an absorbance of 0.7 ± 0.002 at 515 nm. Then, 100 µL of pepper extract obtained with MeOH:H_2_O (80:20) was added to 3.9 mL of the DPPH solution with adjusted absorbance (0.7 Abs); the mixture was stirred and allowed to stand for 30 min for subsequent reading in the spectrophotometer at 515 nm. The percentage of DPPH was calculated using:(1)$$\%DPPH=\left(\frac{A_{control}-A_{Sample}}{A_{control}}\right)\times 100$$
where *A_Control_* is the absorbance of the control (0.7 Abs) and *A_Sample_* is the absorbance of the sample.

### 4.5. Antioxidant Activity by ABTS Assay

The experiments for the determination of the antioxidant activity by the ABTS method were performed according to Re et al. [41] with small modifications. The ABTS substrate working solution was prepared adding 25 µL of 3% hydrogen peroxide solution to 10 mL of ABTS substrate solution. The test sample was prepared mixing 30 µL of the pepper extract (previously diluted 1:50 with methanol) with 60 µL of myoglobin working solution and 450 µL of the ABTS substrate working solution, then left to stand for 5 min at room temperature. Next, 300 µL of stop solution was added and the mixture was allowed to stand at room temperature for one hour; finally, the sample was measured at 405 nm in the spectrophotometer. Trolox, at different concentration (0.015, 0.045, 0.105, 0.21 and 0.42 mM), was used as a standard to determine the antioxidant activity in Habanero pepper samples, expressed in Trolox units.

### 4.6. Analysis of Total Polyphenols

Total polyphenols were quantified using the Folin Ciocalteu colorimetric method reported by Singleton et al. [42] with some modifications. Briefly, 25 µL of pepper extract was mixed with 25 µL of water, followed by an addition of 3 mL of deionized water and 250 µL of Folin Ciocalteu compound, allowed to stand for 5 min. Then, 750 µL of 20% Na_2_CO_3_ and 950 µL of deionized water were added, stirred and allowed to stand for 30 min at room temperature. After 30 min, the absorbance was measured at 765 nm in a spectrophotometer. Gallic acid at different concentrations (5, 10, 15, 20, 25, 30, 40, 60, 80 and 100 µg mL^−1^) was used as a standard to determine the total polyphenols in Habanero pepper samples; results were expressed as mg of gallic acid in 100 g of dried pepper.

### 4.7. Quantification of Polyphenols by UPLC-DAD

Quantification of polyphenols was conducted with a UPLC Acquity H Class (Waters, Milford, MA, USA) with diode array detector (DAD). The column was an Acquity UPLC HSS C18 (100 A°, 1.8 mm, 2.1 × 50 mm) (Waters, Milford, MA, USA). Chromatographic conditions were a flow speed of 0.5 mL min^−1^ with a column temperature set at 45 °C and injection volume of 2 µL. The mobile phases were acetic acid (0.2%) as solvent A and acetonitrile with acetic acid (0.1%) as solvent B. The elution gradient was as follows: 0–10 min from 1% B to 30% B; 10–12 min 30% B; 12–15 min from 30% B to 1% B. The polyphenols peaks measured correspond to the DAD signals at 280 nm.

The calibration curve was prepared with 20 polyphenol standards (gallic acid, protocatechuic acid, catechine, chlorogenic acid, coumaric acid, cinnamic acid, rutin, luteolin, quercetin, kaempferol, vanillin, diosmin, hesperidin, neohesperidin, naringenin, apigenin, diosmetin, vanillic acid, ferulic acid and ellagic acid), purchased from Sigma-Aldrich^®^ (St. Louis, MO, USA). First, a stock solution at a concentration of 1 mg mL^−1^ was prepared from all standards; then, the calibration curve was prepared in the range of 1 to 75 µg mL^−1^. The polyphenols were identified in the samples with the comparison of the retention time of the standards.

### 4.8. Statistical Analysis

Analysis of variance (ANOVA) and minimum significant difference (MSD) at *p* < 0.05 were conducted to test significant differences in polyphenols at five harvesting times. The results presented in Table 1 were evaluated by descriptive and dispersion statistics, being the values presenting the mean ± standard deviation; the statistical test used for separation of means was the last significant difference with 95% confidence level. Linear correlation analysis was performed between the concentration of the different polyphenols and the antioxidant activity by ABTS assay and DPPH radical scavenging. All the statistical analyses were obtained with the software Statgraphics Centurion XVII.II-X64 (Statgraphics Technologies Inc. Virgin, UT, USA).

## 5. Conclusions

The results indicate that grade of maturity, harvest time (PTD) and the interaction between these factors have a significant effect in almost all the polyphenols evaluated. The concentration of the majority of polyphenols, such as catechin, chlorogenic acid, vanillic acid and ellagic acid, decreased through the harvest days; in contrast, an increase of the content of the other polyphenols, such as gallic and protocatechuic acid, was evidenced. The polyphenols with the highest concentration values were catechin (355.30 ± 5.81 mg/100 g) and chlorogenic acid (79.97 ± 2.02 mg/100 g) in mature Habanero peppers harvested at 160 PTD. Catechin and vanillic acid were the only polyphenols that presented a good fit (R^2^ > 0.7) in immature and mature Habanero peppers correlations with both antioxidant activity methods utilized; this same behavior was also observed with the total polyphenols. The linear correlation analysis indicated that the antioxidant activity is strongly influenced by the presence of polyphenols. The polyphenol that more influenced the antioxidant activity is the catechin in Habanero peppers with a good linear correlation with the antioxidant activity and also identified as the major polyphenol. This allows to conclude that antioxidant activity decreases through harvest times, decreasing also the content of the main polyphenols in *Capsicum chinense* fruits. Indicating the results obtained that the peppers harvested in the first PTD presented the highest antioxidant activity and concentration of total polyphenols, with 160 PTD having the best results of the five evaluated dates. This knowledge could be used to select a harvest time that is most favorable for obtaining valuable Habanero peppers with a high content of polyphenols, for which there is a commercial interest in the functional food sector.

## Figures and Tables

**Figure 1 plants-09-01394-f001:**
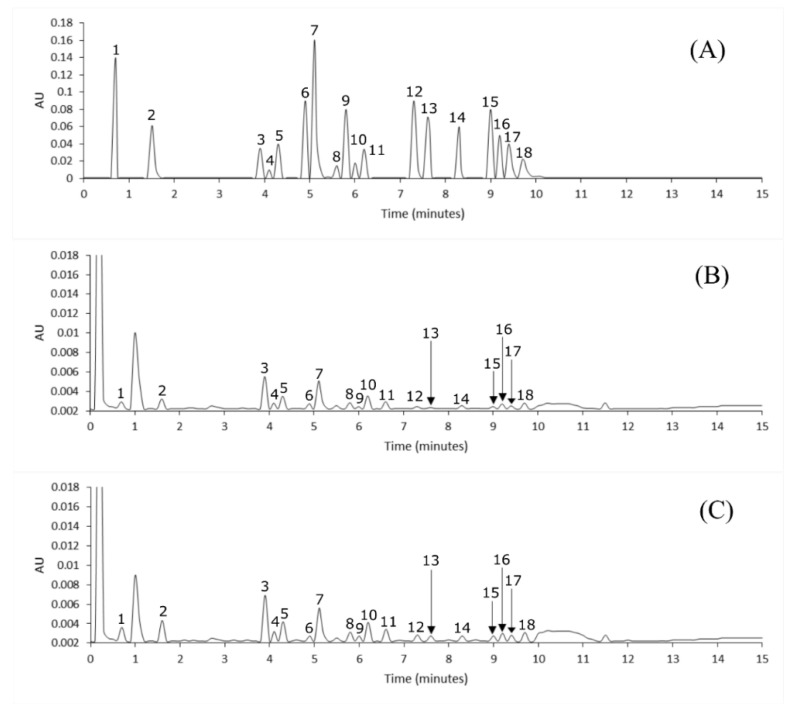
Chromatogram of: (**A**) Polyphenols standards at a concentration of 75 µg mL^−1^ analyzed at 280 nm. (**B**) Sample of immature Habanero pepper harvested at 160 post-transplant days (PTD). (**C**) Sample of mature Habanero pepper harvested at 160 PTD at 280 nm. The numbers correspond to: (1) gallic acid, (2) protocatechuic acid, (3) catechine, (4) vanillic acid, (5) chlorogenic acid, (6) vanillin, (7) coumaric acid, (8) ferulic acid, (9) cinnamic acid, (10) ellagic acid, (11) rutin, (12) diosmin + hesperidin, (13) neohesperidin, (14) luteolin + quercetin, (15) naringenin, (16) apigenin, (17) kaempferol and (18) diosmetin.

**Figure 2 plants-09-01394-f002:**
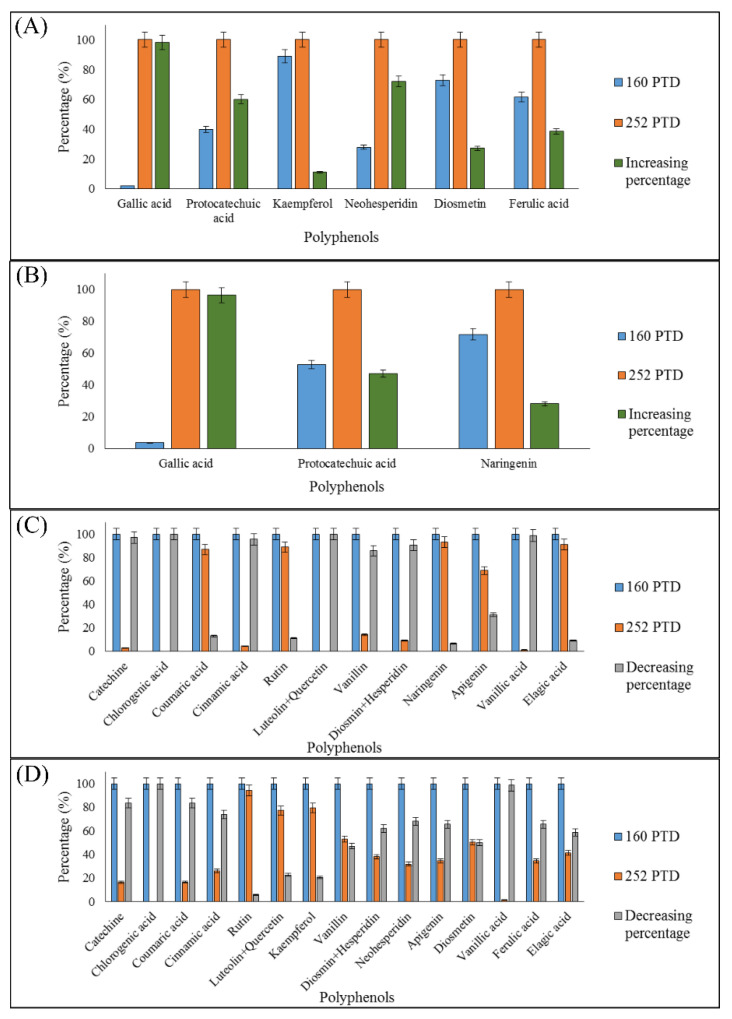
Change in the percentage of polyphenols between the first (160 PTD) and the last harvest (252 PTD). (**A**) Polyphenols from immature peppers that increased during harvests. (**B**) Polyphenols from mature peppers that increased during harvests. (**C**) Polyphenols from immature peppers that decreased during harvests. (**D**) Polyphenols from mature peppers that decreased during harvests.

**Table 1 plants-09-01394-t001:** Results of the quantification of polyphenols in mature (orange) and immature (green) Habanero pepper (*Capsicum chinense* Jacq.) at different harvest times (or PTD).

Polyphenol (mg/100 g of Dry Pepper)	160 PTD		209 PTD		223 PTD		237 PTD		252 PTD	
	Green	Orange	Green	Orange	Green	Orange	Green	Orange	Green	Orange
Gallic acid	0.56 ± 0.2 ^c^	1.73 ± 0.45 ^B^	0.00 ± 0.00 ^d^	0.00 ± 0.00 ^C^	0.00 ± 0.00 ^d^	0.00 ± 0.00 ^C^	6.93 ± 1.09 ^b^	0.00 ± 0.00 ^C^	29.45 ± 10.11 ^a^	48.40 ± 12.31 ^A^
Protocatechuic acid	15.91 ± 0.19 ^b^	85.75 ± 10.45 ^C^	41.94 ± 0.53 ^a^	102.85 ± 14.57 ^C^	6.04 ± 0.02 ^c^	122.73 ± 3.31 ^B^	5.67 ± 0.23 ^d^	56.22 ± 9.89 ^D^	39.84 ± 12.44 ^a^	162.21 ± 0.55 ^A^
Catechine	182.52 ± 2.74 ^a^	355.30 ± 5.81 ^A^	117.42 ± 1.91 ^b^	319.73 ± 3.19 ^B^	3.79 ± 0.17 ^d^	53.36 ± 4.72 ^D^	3.46 ± 0.14 ^d^	17.99 ± 0.27 ^E^	5.11 ± 0.03 ^c^	58.33 ± 0.02 ^C^
Chlorogenic acid	6.86 ± 0.20 ^b^	79.97 ± 2.02 ^A^	28.36 ± 0.29 ^a^	69.5 ± 0.1 ^B^	0.33 ± 0.01 ^d^	1.41 ± 0.38 ^C^	0.74 ± 0.05 ^c^	0.40 ± 0.11 ^D^	0.00 ± 0.00 ^e^	0.00 ± 0.00 ^E^
Coumaric acid	0.78 ± 0.15 ^a^	2.31 ± 0.54 ^A^	0.05 ± 0.01 ^b^	0.58 ± 1.2 ^B^	0.00 ± 0.00 ^c^	0.45 ± 0.21 ^B^	0.00 ± 0.00 ^c^	0.00 ± 0.00	0.67 ± 0.05 ^a^	0.38 ± 0.09 ^B^
Cinnamic acid	7.81 ± 0.16 ^a^	25.78 ± 5.14 ^A^	1.04 ± 0.01 ^c^	8.07 ± 2.71 ^B^	5.92 ± 0.87 ^b^	0.42 ± 0.06 ^C^	0.48 ± 0.17 ^d^	0.24 ± 0.04 ^D^	0.35 ± 0.02 ^d^	6.73 ± 0.97 ^B^
Rutin	19.19 ± 8.52 ^a^	46.05 ± 5.57 ^A^	6.41 ± 0.69 ^c^	35.11 ± 15.73 ^A^	17.25 ± 6.59 ^ab^	3.64 ± 0.04 ^C^	1.45 ± 0.04 ^d^	10.95 ± 0.09 ^B^	17.06 ± 0.75 ^b^	43.41 ± 4.86 ^A^
Luteolin + quercetin	0.65 ± 0.17 ^a^	1.38 ± 0.09 ^B^	0.00 ± 0.00 ^c^	0.00 ± 0.00 ^D^	0.00 ± 0.00 ^c^	18.99 ± 4.62 ^A^	0.41 ± 0.18 ^b^	17.19 ± 8.41 ^A^	0.00 ± 0.00 ^c^	1.07 ± 0.06 ^C^
Kaempferol	0.44 ± 0.09 ^a^	1.87 ± 0.01 ^A^	0.00 ± 0.00 ^c^	0.00 ± 0.00 ^B^	0.06 ± 0.04 ^b^	0.00 ± 0.00 ^B^	0.00 ± 0.00 ^c^	0.00 ± 0.00 ^B^	0.49 ± 0.06 ^a^	1.49 ± 0.91 ^A^
Vanillin	5.86 ± 0.05 ^a^	2.07 ± 0.09 ^A^	1.36 ± 0.04 ^b^	1.32 ± 0.02 ^B^	0.16 ± 0.01 ^c^	0.19 ± 0.02 ^D^	0.19 ± 0.07 ^c^	0.20 ± 0.09 ^D^	0.82 ± 0.75 ^bc^	1.09 ± 0.02 ^C^
Diosmin + hesperidin	6.53 ± 0.09 ^a^	14.28 ± 0.07 ^A^	1.29 ± 0.01 ^c^	2.67 ± 0.29 ^C^	4.98 ± 0.74 ^b^	0.85 ± 0.01 ^D^	0.72 ± 0.02 ^d^	0.82 ± 0.11 ^D^	0.60 ± 0.01 ^e^	5.44 ± 0.05 ^B^
Neohesperidin	0.22 ± 0.01 ^c^	2.65 ± 0.03 ^A^	0.21 ± 0.01 ^c^	0.49 ± 0.36 ^D^	0.00 ± 0.00 ^d^	2.03 ± 0.55 ^B^	0.29 ± 0.01 ^b^	2.44 ± 0.03 ^B^	0.79 ± 0.02 ^a^	0.85 ± 0.06 ^C^
Naringenin	0.38 ± 0.11 ^ab^	0.49 ± 0.23 ^D^	0.34 ± 0.03 ^b^	1.89 ± 0.01 ^C^	0.39 ± 0.02 ^a^	5.51 ± 2.46 ^B^	0.34 ± 0.01 ^b^	12.32 ± 2.22 ^A^	0.35 ± 0.03 ^b^	0.69 ± 0.26 ^D^
Apigenin	0.49 ± 0.04 ^a^	2.05 ± 0.08 ^A^	0.41 ± 0.01 ^b^	1.11 ± 0.01 ^B^	0.31 ± 0.01 ^c^	0.91 ± 0.26 ^BC^	0.19 ± 0.09 ^d^	1.21 ± 0.20 ^B^	0.34 ± 0.04 ^c^	0.79 ± 0.03 ^C^
Diosmetin	0.59 ± 0.13 ^b^	1.19 ± 0.85 ^BC^	0.64 ± 0.09 ^b^	2.53 ± 0.02 ^A^	0.76 ± 0.02 ^a^	0.68 ± 0.01 ^BC^	0.57 ± 0.09 ^b^	0.61 ± 0.01 ^BD^	0.81 ± 0.31 ^ab^	0.60 ± 0.01 ^BD^
Vanillic acid	25.24 ± 0.13 ^a^	45.48 ± 0.55 ^A^	22.41 ± 0.23 ^b^	11.81 ± 9.86 ^B^	0.39 ± 0.02 ^d^	1.06 ± 0.05 ^C^	0.51 ± 0.02 ^c^	0.35 ± 0.26 ^E^	0.31 ± 0.01 ^e^	0.65 ± 0.02 ^D^
Ferulic acid	2.52 ± 0.16 ^b^	12.12 ± 0.05 ^A^	1.92 ± 0.15 ^c^	3.60 ± 0.77 ^B^	0.05 ± 0.04 ^e^	0.25 ± 0.03 ^C^	1.46 ± 0.04 ^d^	0.45 ± 0.39 ^C^	4.09 ± 0.18 ^a^	4.17 ± 0.17 ^B^
Ellagic acid	3.67 ± 0.02 ^a^	7.78 ± 0.03 ^A^	1.79 ± 0.36 ^b^	5.44 ± 0.81 ^B^	1.99 ± 0.17 ^b^	2.58 ± 0.02 ^C^	1.69 ± 0.03 ^b^	2.34 ± 0.53 ^C^	3.34 ± 0.39 ^a^	3.22 ± 0.93 ^C^
Total polyphenols	79.34 ± 0.42 ^a^	148.77 ± 2.57 ^A^	68.38 ± 1.39 ^b^	127.00 ± 0.46 ^B^	50.38 ± 0.26 ^c^	97.44 ± 0.31 ^C^	42.07 ± 0.46 ^d^	100.84 ± 0.52 ^D^	39.61 ± 0.26 ^e^	87.08 ± 0.31 ^E^
DPPH (%)	87.50 ± 0.51 ^a^	89.50 ± 0.30 ^A^	86.79 ± 0.51 ^a^	88.29 ± 0.61 ^B^	85.93 ± 0.10 ^b^	86.43 ± 0.81 ^C^	85.57 ± 0.40 ^bc^	86.21 ± 0.30 ^C^	84.71 ± 0.81 ^c^	85.86 ± 0.61 ^C^
ABTS (mg of trolox/g)	24.47 ± 0.18 ^a^	24.96 ± 0.07 ^A^	23.24 ± 0.38 ^b^	23.81 ± 0.19 ^B^	22.51 ± 0.41 ^bc^	22.68 ± 0.09 ^C^	21.69 ± 0.30 ^c^	22.42 ± 0.07 ^D^	22.54 ± 0.38 ^bc^	22.73 ± 0.22 ^C^

Note: Values presented are means of polyphenols ± standard deviation. Different letters in the same row indicate statistically significant differences using a least significant difference (LSD) test at *p* ≤ 0.05. Lowercase letters are for comparing immature peppers, while uppercase letters are for comparing mature peppers.

**Table 2 plants-09-01394-t002:** *P* values of the different factors evaluated and their respective interactions for each quantified polyphenols, total polyphenols and antioxidant activity by DPPH (2,2-Diphenyl-1-picrylhydrazyl) and ABTS (2,2′-azino-di-3-ethylbenzthiazoline sulfonic acid).

Polyphenol	A: PTD	B: Maturity	AB
Gallic acid	<0.0001 *	0.2752	0.0464 *
Protocatechuic acid	<0.0001 *	<0.0001 *	0.0001 *
Catechin	<0.0001 *	<0.0001 *	<0.0001 *
Chlorogenic acid	<0.0001 *	<0.0001 *	<0.0001 *
Coumaric acid	<0.0001 *	0.0005 *	0.0006 *
Cinnamic acid	0.0012 *	0.0150	0.0169 *
Rutin	0.2381	0.0853	0.4596
Luteolin + quercetin	0.2415	0.0474 *	0.2404
Kaempferol	0.0002 *	0.0045 *	0.0115 *
Vanillin	<0.0001 *	0.0001 *	<0.0001 *
Diosmin + hesperidin	0.0009 *	0.0744	0.0308 *
Neohesperidin	0.3775	0.0034 *	0.1838
Naringenin	0.0001 *	<0.0001 *	0.0001 *
Apigenin	<0.0001 *	<0.0001 *	0.0002 *
Diosmetin	0.0048 *	0.0062 *	0.0025 *
Vanillic acid	<0.0001 *	0.1641	0.0006 *
Ferulic acid	<0.0001 *	<0.0001 *	<0.0001 *
Ellagic acid	<0.0001 *	<0.0001 *	0.0001 *
Total polyphenols	<0.0001 *	<0.0001 *	<0.0001 *
DPPH	<0.0001 *	<0.0001 *	0.0286 *
ABTS	<0.0001 *	0.0001 *	0.2258

Note: (*) = Significant effect; PTD = post-transplant days; AB = Interaction of maturity with post-transplant days.

**Table 3 plants-09-01394-t003:** Correlation of the concentration of polyphenols with the antioxidant activity determined by DPPH radical scavenging and ABTS assay in *Capsicum chinense*, and the equation of the linear correlation was obtained.

Polyphenol	DPPH				ABTS			
	Immature (Green Color)		Mature (Orange Color)		Immature (Green Color)		Mature (Orange Color)	
	r^2^	Equation	r^2^	Equation	r^2^	Equation	r^2^	Equation
Gallic acid	0.6099	y = −0.068x + 86.605	0.1797	y = −0.032x + 87.573	0.1184	y = −0.029x + 23.105	0.0628	y = −0.012x + 23.425
Protocatechuic acid	0.0248	y = −0.010x + 86.317	0.1002	y = −0.013x + 88.607	0.0416	y = 0.012x + 22.620	0.0411	y = −0.005x + 23.878
Catechin	0.7245	y = 0.012x + 85.362	0.8661	y = −0.009x + 85.756	0.7842	y = 0.012x + 22.150	0.8989	y = 0.006x + 22.278
Chlorogenic acid	0.2587	y = 0.048x + 85.749	0.8794	y = 0.037x + 86.124	0.1533	y = 0.036x + 22.629	0.8934	y = 0.025x + 22.537
Coumaric acid	0.0176	y = 0.389x + 85.984	0.7168	y = 1.493x + 86.146	0.2638	y = 1.449x + 22.457	0.8172	y = 1.066x + 22.507
Cinnamic acid	0.2074	y = 0.115x + 85.744	0.7139	y = 0.126x + 86.217	0.3845	y = 0.150x + 22.422	0.8344	y = 0.091x + 22.549
Rutin	0.0004	y = −0.002x + 86.128	0.1803	y = 0.025x + 86.550	0.2362	y = 0.052x + 22.258	0.2301	y = 0.019x + 22.766
Luteolin + quercetin	0.2027	y = 1.412x + 85.802	0.1828	y = −0.055x + 87.683	0.1309	y = 1.095x + 22.659	0.2670	y = −0.045x + 23.645
Kaempferol	0.0009	y = −0.142x + 86.128	0.1395	y = 0.618x + 86.842	0.1546	y = 1.747x + 22.541	0.3022	y = 0.608x + 22.892
Vanillin	0.4656	y = 0.328x + 85.551	0.6183	y = 1.605x + 85.690	0.7537	y = 0.402x + 22.216	0.8406	y = 1.252x + 22.079
Diosmin + hesperidin	0.1993	y = 0.139x + 85.707	0.4842	y = 0.202x + 86.287	0.3772	y = 0.185x + 22.369	0.7147	y = 0.164x + 22.512
Neohesperidin	0.3551	y = −2.317x + 86.805	0.0002	y = −0.017x + 87.286	0.0619	y = −0.933x + 23.174	0.0064	y = 0.065x + 23.190
Naringenin	0.0870	y = 12.681x + 81.540	0.2513	y = −0.160x + 87.925	0.1122	y = 13.894x + 17.894	0.3793	y = −0.131x + 23.849
Apigenin	0.2022	y = 3.492x + 84.884	0.6610	y = 2.455x + 84.316	0.4363	y = 4.948x + 21.166	0.6610	y = 1.702x + 21.262
Diosmetin	0.3381	y = −4.178x + 88.921	0.2963	y = 1.012x + 86.120	0.0026	y = −0.355x + 23.130	0.2455	y = 0.616x + 22.608
Vanillic acid	0.7061	y = 0.075x + 85.364	0.7226	y = 0.070x + 86.424	0.7017	y = 0.072x + 22.182	0.8980	y = 0.052x + 22.679
Ferulic acid	0.0447	y = −0.165x + 86.431	0.6073	y = 0.263x + 86.173	0.0162	y = 0.096x + 22.698	0.8143	y = 0.204x + 22.461
Ellagic acid	0.0174	y = 0.160x + 85.702	0.8976	y = 0.654x + 84.466	0.2591	y = 0.594x + 21.409	0.9474	y = 0.449x + 21.383
Total polyphenols	0.8348	y = 0.0160x + 82.648	0.8999	y = 0.0612x + 80.385	0.8191	y = 0.0589x + 19.592	0.8922	y = 0.0407x + 18.726

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
