# Peer review of "Polyphenols Content in Capsicum chinense Fruits at Different Harvest Times and Their Correlation with the Antioxidant Activity"

_plants, 2020, doi:10.3390/plants9101394_

Round 1

Reviewer 1 Report

The manuscript is very well written. The introduction introduces the reader to the subject well. The methods and materials are well chosen and described with one exception. Description and presentation of results in tables and figures do not arouse any reservations. In the discussion, the authors explain in detail the correlation relationships based on well-selected literature. Nevertheless, I have a few minor comments and one more important. List of comments:

Line:

26-28              In my opinion, these sentences are redundant, because the article, apart from its scientific meaning, is also of great importance for agricultural practice and it would be better to summarize with specific conclusions which time of harvesting peppers are the most favorable and for what reason.

29                    In my opinion, keywords should’n repeat with the words in the title and in the abstract (wider search - better quoting).

65                    Bee propolis? Please explain.

198-199          This sentence is most likely incomplete.

216                  There are 49 days between the first harvest time (160 PTD) and the second (209 PTD), and there is approximately 15 days off between harvesting times. Could you explain why?

219-220          Is it the anti-trips insect screens?

220                  12 kg dry or wet of soli? If wet, what was the humidity of soil?

226                  Please specify the composition of the fertilizer Triple 18 Ultrasol?

234                  “green or immature and orange or mature” - This is incomprehensible - Please explain.

237                  Please indicate how much peppers was ground (sample size in gram)?

268                  “Folic Ciocalteu”...? There shouldn't be "Folin ..."?

In the methodology, I did not find a description of the light, thermal and humidity conditions. During the harvest after 160 PTD, were the day length and temperature the same as for example after 252 PTD? Was there any artificial lighting or regulation of humidity and temperature? this is a major shortcoming as these conditions could largely determine the level of the parameters tested, especially in greenhouse cultivation. On the basis of these data, it is possible to determine the most favorable lighting, humidity and temperature conditions for obtaining the most valuable pepper.

Similarly, in the discussion and conclusions, in my opinion, the authors should refer relationship between the harvest date and changes in meteorological conditions (especially light and temperature). In the conclusions (as in the abstract), in my opinion, there is missing conclusion as to which harvest term is the most favorable for obtaining the most valuable habanero pepper.

Best regards

Reviewer 2 Report

The manuscript by Oney-Montalvo et al reports on polyphenols content in Capsicum chinense fruits at different harvest times. In addition, authors try to provide information on the correlation between antioxidant activity and the total/ individual polyphenolic content.

The reported results on polyphenols as antioxidant agents doesn't add much to the knowledge, and I believe that the manuscript couldn’t give a substantial contribution to the field.

Some sentences are incomprehensible:

1) ’These results can be associated to an increase in the biosynthesis of polyphenols caused by the process of maturation and by the accumulation of NO3 and PO4’ What are NO3 and PO4?????

2) Previous work conducted by Dubey et al. [11] determined catechin (18.03 mg 100g-1) as the main polyphenol in peppers, followed by quercetin (3.87 mg 100g-1), protocatechic acid (2.06 mg 100g-1) and rutin (0.82 mg 100g-1). This results coincided with the reported by Troconis-Torres et al. [12] that quantified catechin (52.25 mg 100g-1) as the major polyphenol in habanero pepper’. I don’t understand how these concentrations have been expressed.

3) Linear correlation or lineal correlation?

4) protocatechic acid? Are you sure?

5) Table 3 is not clear

6) Furthermore, it is necessary to include UPLC chromatograms as supporting information.

The manuscript is not carefully written; a great number of typographical and grammatical errors as well as many inaccuracies are present throughout the text. The manuscript is written in poor English and professional English editing is highly recommended.

The manuscript doesn’t reach the standard of this journal and thus, in my opinion, it is not acceptable in this form.

Reviewer 3 Report

In this work, soils and fertilizers were used, as well as a large dose of phytoregulators and fertilizers, without any previous analysis having been made to the soils used. You should have knowledge of the composition of the soil, pH, etc., you have been using a specific soil, but have not tested it, where information is lacking. They should also have post-fertilization analyzes to get a sense of the plant's physiological development. The application of plant hormones interferes with the development of the primary, but also secondary, metabolism of the plant, by applying plant hormones they may be profoundly modifying the production of polyphenols, as well as other compounds of the secondary metabolism, making it difficult to analyze the true chemical profile of this species as well as making any kind of comparison between species in Mexico and India.

Round 2

Reviewer 2 Report

It was already very clear that concentration is expressed in milligrams of the polyphenol for 100 grams of dry pepper but the way that authors write it is wrong! For example, ‘catechin (18.03 mg 100g-1 of dry pepper)’ it has no meaning and considering that the hint was not taken, I suggest replacing ‘18.03 mg 100g-1 of dry pepper’ with ‘18.03 mg/ 100g of dry pepper’. English language is an issue that has not yet been addressed, in the text there are the same grammatical errors. For example, see ‘This results’. In addition, in the first revision I ask ‘Furthermore, it is necessary to include UPLC chromatograms as supporting information’ but the issue was not dealt with in the response. Definitively, for these reasons in my opinion the manuscript is not yet acceptable in this form.

Reviewer 3 Report

the depth of the work remains low, giving rise to little originality, however the authors made an effort to improve the points marked as weak. In this context, I believe that the paper can be edited, inserted in a deeper and broader work with botanical and edaphoclimatic data.

Round 3

Reviewer 2 Report

The authors revised the manuscript according to suggested comments therefore, in my opinion, it is now suitable for publication on Plants in present form.

Reviewer 3 Report

Although there are still some issues, the relevance of them does not seem to me that the publication of this work